# Prioritizing disease-related rare variants by integrating gene expression data

Hanmin Guo [1,2], Alexander Eckehart Urban[2,3]*, Wing Hung Wong[1,4]*

**1** Department of Statistics, Stanford University, Stanford, California, United States of America, **2** Department of Psychiatry and Behavioral Sciences, Stanford University School of Medicine, Stanford, California, United States of America, **3** Department of Genetics, Stanford University School of Medicine, Stanford, California, United States of America, **4** Department of Biomedical Data Science, Stanford University School of Medicine, Stanford, California, United States of America

* aeurban@stanford.edu (AEU); whwong@stanford.edu (WHW)

## Abstract

Rare variants, comprising the vast majority of human genetic variations, are likely to have more deleterious impact in the context of human diseases compared to common variants. Here we present carrier statistic, a statistical framework to prioritize disease-related rare variants by integrating gene expression data. By quantifying the impact of rare variants on gene expression, carrier statistic can prioritize those rare variants that have large functional consequence in the patients. Through simulation studies and analyzing real multi-omics dataset, we demonstrated that carrier statistic is applicable in studies with limited sample size (a few hundreds) and achieves substantially higher sensitivity than existing rare variants association methods. Application to Alzheimer's disease reveals 16 rare variants within 15 genes with extreme carrier statistics. We also found strong excess of rare variants among the top prioritized genes in patients compared to that in healthy individuals. The carrier statistic method can be applied to various rare variant types and is adaptable to other omics data modalities, offering a powerful tool for investigating the molecular mechanisms underlying complex diseases.

## Author summary

Existing rare variants association methods often lack statistical power when sample sizes are small. Here we propose a novel integrative statistical framework, the carrier statistic, which can leverage paired genotype and gene expression data to quantify the functional impact of rare variants and enhance detection power of rare variants responsible for disease. Extensive simulations demonstrate that carrier statistic provides well-calibrated false discovery rates, shows substantially higher sensitivity compared to existing methods, and remains robust under unbalanced case-control ratios. Through analyzing real multi-omics dataset for Alzheimer's disease, we identified 16 rare variants within 15 genes with extreme carrier statistics. We hope that the results presented in this paper can highlight the promise of the carrier statistic approach and will encourage future disease studies to collect both genotype and gene expression data for the same individuals. As multi-omics

**Data Availability Statement:** Carrier statistic tool is available at https://github.com/SUwonglab/carrier-stat. The WGS data in the Whole Genome Sequence Harmonization Study (https://www.synapse.org/#!Synapse:syn22264775) and the

RNA-seq data in the RNAseq Harmonization Study (https://www.synapse.org/#!Synapse: syn21241740) are publicly available on the AD Knowledge Portal platform through completion of a data use certificate. The gnomAD v2.1.1 data consisting of 125,748 exomes were downloaded from https://gnomad.broadinstitute.org/. The gene expression data (gene reads count) from GTEx project version 8 were downloaded from the GTEx Portal, https://gtexportal.org/home/downloads/ adult-gtex/bulk_tissue_expression. SAIGE-GENE+, https://saigegit.github.io/SAIGE-doc/. coloc, https:// chr1swallace.github.io/coloc/. SKAT-O, https:// cran.rproject.org/web/packages/SKAT/.

**Funding:** This work was partially supported by the NIH (R01 HG010359 to W.H.W; R01 MH116529 to A.E.U.; and P50 HG007735 to W.H.W. and A.E.U.) and the NSF DMS (2310788 to W.H.W.). The funders had no role in study design, data collection and analysis, decision to publish, or preparation of the manuscript.

**Competing interests:** The authors have declared that no competing interests exist.

and genome sequencing data continue to expand, we anticipate that carrier statistics will be a valuable tool for elucidating the molecular mechanism underlying human complex diseases.

## Introduction

Rare variants (minor allele frequency (MAF) < 1%) constitute the vast majority of human genetic variations [1,2]. They are on average more deleterious compared with common variants, and thus undergo stronger selection and remain at low frequency in the general population. By analyzing large cohorts of whole genome sequencing (WGS) and whole exome sequencing (WES) data, researchers have identified some rare variants-trait associations [3–6] and have shown that rare variants contribute to a large proportion of missing heritability that cannot be explained by common variants [7].

Rare variants on average confer larger effects on gene expression and complex diseases and are easier to map to causal genes than common variants [8]. However, statistical power to identify disease-associated rare variants, especially for ultra-rare variants or even singletons, is limited, given that the sample size is not especially high or the effect size is not too large. Variants collapsing methods (burden test, variance component test, omnibus test) are proposed to circumvent this obstacle [6,9–11], which evaluate association for multiple variants in a biologically relevant region, such as a gene, instead of testing the effect of single variant. These methods work well under the assumption that multiple variants in a gene cumulatively contribute to the disease risk with each individual allele explaining only a small fraction of the cases, but resolution to pinpoint the risk variants may be diluted when the assumption is violated. Recent studies based on large scale WES have revealed rare protein truncating variants associated with a wide range of phenotypes [4,5,12,13]. These variants exert extreme effects on the function of genes and their encoded proteins, underscoring the importance of considering how rare variants are related to gene expression.

Complementary to genome sequencing assays, RNA-seq can quantitatively measure gene expression level and provide molecular cause of complex diseases, especially rare diseases. Previous studies have shown that rare variants are enriched near genes with aberrant gene expression [14–16]. We posit that those rare variants will be more prone to have effect on disease pathology. In this work, we propose carrier statistic, a statistical framework for prioritizing those rare variants with large functional consequence in patients by integrating gene expression data. We demonstrate superior performance of our method through extensive simulations and an application study to Alzheimer's disease, where given a limited sample size, existing rare variants association methods without functional gene expression data alone cannot provide positive findings.

## Results

### Method overview

Our method stems from the expectation that diseased population shows enrichment in rare variants that have large impact on expression for disease-related genes. Suppose we have genotypes (e.g. variants call from WGS data) and gene expression measurements (e.g. reads count from RNA-seq data) on a disease relevant cell type for both patients and healthy controls. For each rare variant-gene pair, we calculate the expression association z-score for rare variant carriers by using the gene expression from individuals without the variant as the null distribution

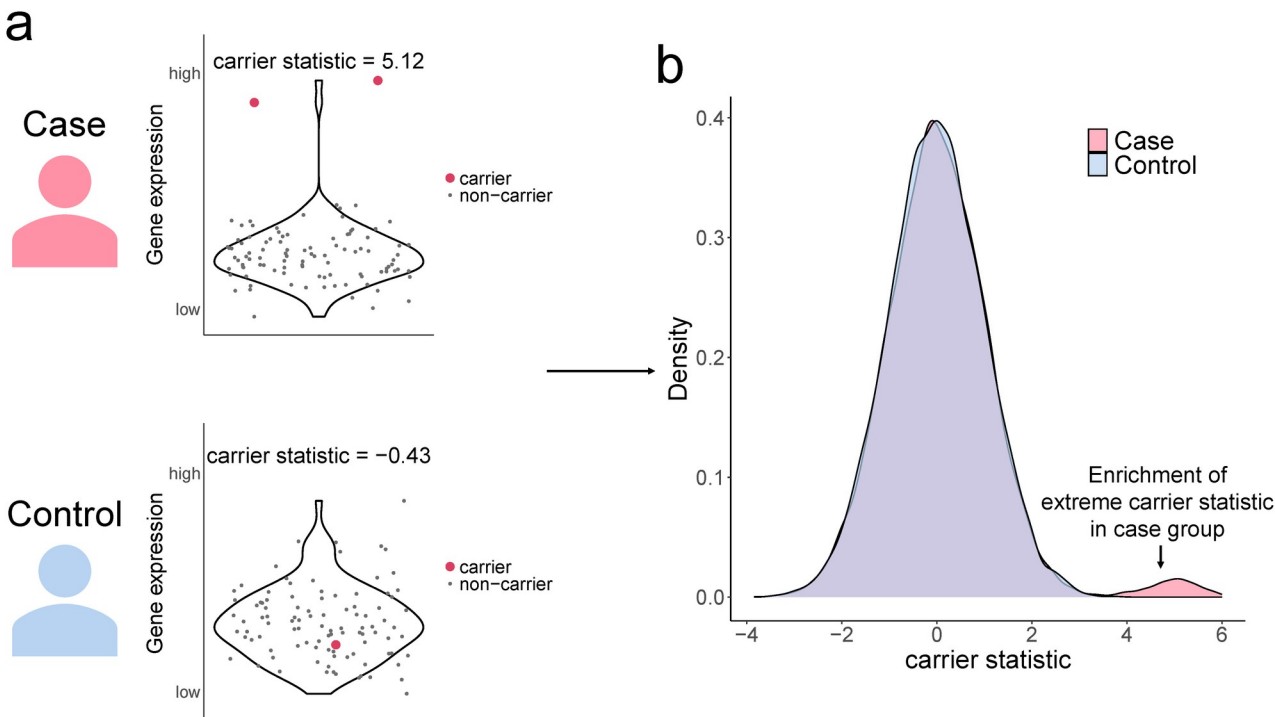

**Fig 1. The distribution of carrier statistics in the case group exhibits a heavier tail compared to that in the control group.** (a). Violin plots show gene expression level among carriers and non-carriers of a rare variant, with each data point jittered along the x-axis to improve visualization. (b). Distribution of carrier statistics in the case group and control group.

(**Fig 1A**). The expression association z-score, which we term as carrier statistic, is calculated separately within the case and the control group. We only consider rare variant-gene pairs wherein the variant is located within the exonic part of the gene throughout this study. The carrier statistic can be interpreted as expression effect size which quantifies the degree to which the rare variant impacts gene expression level. We assume that most rare variants do not have a large impact on gene function, so the distribution of carrier statistic will be centered around 0. Now, if a gene is relevant for the disease, then conditioning on having the disease will bias the sampling towards individuals carrying rare variants with large functional impact on the gene expression. While rare variants that cause extreme expression levels for genes unrelated to the disease may also exist in the control group, they are uncommon. Therefore, the distribution of carrier statistic in the case group will exhibit a heavier tail compared to that in the control group (**Fig 1B**). We prioritize those rare variants and genes with outlier carrier statistic in the case group. False discovery rate (FDR) can be computed as the ratio of tail probability for carrier statistic between two groups (**Methods**).

## Simulation results

We first carried out simulations to assess whether the carrier statistic-based method would produce false positive findings. We simulated genotypes based on whole exome sequencing (WES) data from the Genome Aggregation Database (gnomAD) [2] and simulated gene expression profiles based on RNA-seq data in whole blood tissue from the Genotype-Tissue Expression (GTEx) project (**Methods**). We perturbed the expression level of the causal genes for causal variants carriers by assuming that rare causal variants have large functional impact on disease-related genes. FDR for carrier statistic was well-calibrated in all simulation settings

with varying penetrance of causal variant, prevalence in causal variant noncarriers, and number of causal variants per causal gene (**S1 Fig**). We also checked if using gene expression from noncarriers in two groups together rather than separately as null distribution will produce false positive findings (**Methods**). In this case, we observed substantial inflation in FDR, particularly when patients have systematic changes in their gene expression profile from healthy controls. In contrast, using gene expression from noncarriers in the same group as null distribution consistently gave well-calibrated error rates (**S2 Fig**). Finally, the presence of rare protective variants will not affect the validity or performance of carrier statistic, assuming these variants typically maintain gene expression levels within a normal range rather than causing significant alterations (**Methods** and **S3 Fig**).

Next, based on the simulated data, we benchmarked the performance of carrier statistic with two existing rare variants association methods: SKAT-O [11] and SAIGE-GENE+ [17]. SKAT-O linearly combines burden test statistic [9], which counts the number of rare variants within a gene, and Sequence Kernel Association Test (SKAT) [10], which computes a gene-level variance component score statistic that allows bidirectional effect of different variants. The unified test SKAT-O is preferred over burden test and SKAT when the underlying genetic architecture of the disease is not known. SAIGE-GENE+ [17] is a method designed for variants collapsing association test with unbalanced case-control phenotypes. We also included coloc [18,19], a method that estimates the posterior probability of colocalization between gene expression and GWAS association signals, for comparison. While all four methods successfully controlled FDR at the nominal level (**S1 Fig**), carrier statistic achieved higher sensitivity than the three variants collapsing methods under all simulation settings (**Fig 2**). We believe the low sensitivity of burden-like statistics is due to the small number of case samples that can be attributed to the causal variants in any gene region, which makes it difficult to attain statistical significance of enrichment of rare variants burden for any causal gene region (**S1–S5 Tables**). We also investigated if an unbalanced case-control ratio would affect the performance of our method. We found that carrier statistic maintained well-calibrated FDR and showed substantially higher sensitivity than other three methods in simulations with 500 cases and 50,000 controls (**S4 Fig**). Under this condition, SAIGE-GENE+ and coloc controlled FDR effectively but showed low sensitivity, while a significant proportion of false positives was observed for SKAT-O, a finding also reported in the SAIGE-GENE+ paper [17].

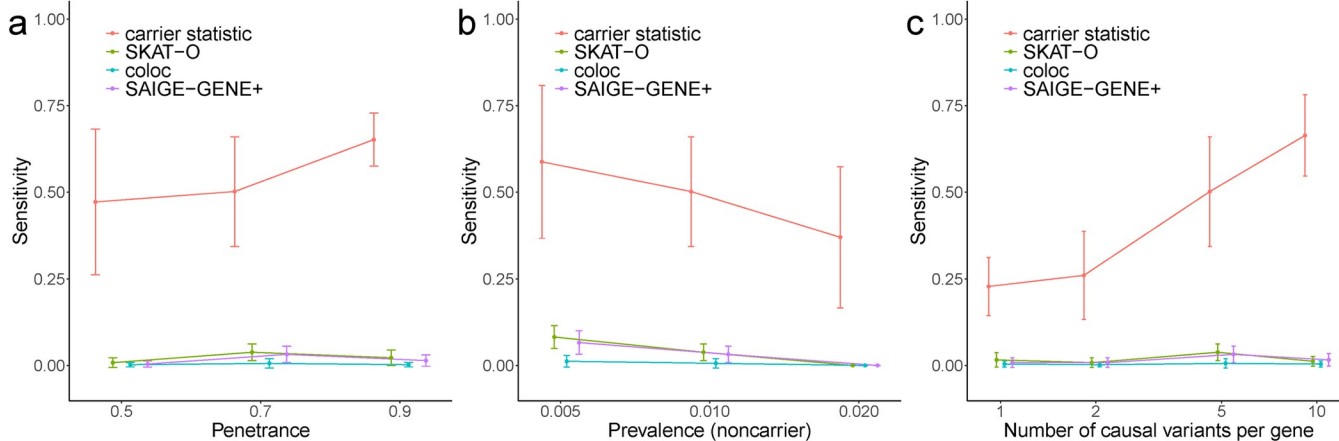

**Fig 2. Carrier statistic achieves higher sensitivity than SKAT-O, coloc, and SAIGE-GENE+ in simulations with varying (a) penetrance of causal variant, (b) prevalence in causal variant noncarriers, and (c) number of causal variants per causal gene.** Error bar shows standard deviation across 100 simulation repeats.

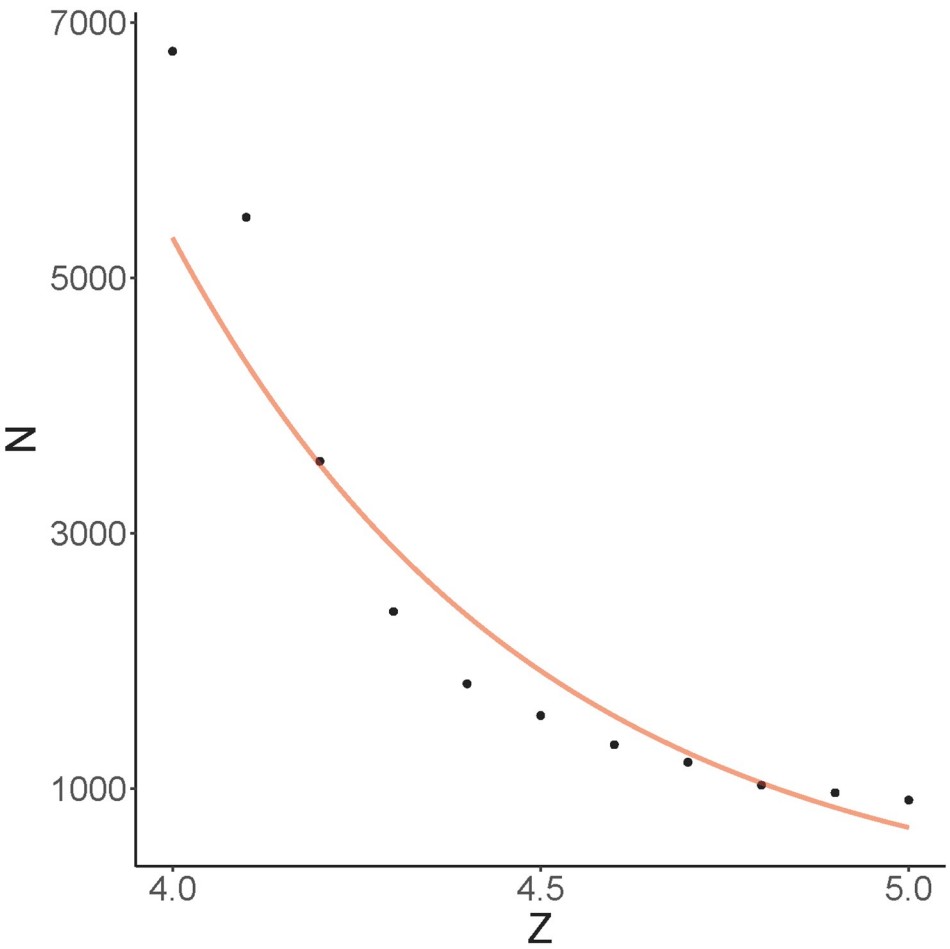

**Fig 3. Required sample size for carrier statistic to attain 80% sensitivity based on simulations.** Y-axis denotes total sample size with 50% case-control ratio and x-axis denotes effect size of causal variant on gene expression. Log-transformed sample size is regressed on Z and the fitted curve is shown in red.

We also performed empirical power analysis, which provides guidance for designing new disease association studies with genome sequencing and RNA-seq data. Simulations were repeated 50 times to determine the sample size required for achieving 80% sensitivity, given that the penetrance of causal variant is 70%, the prevalence of causal variant noncarrier is 1%, and there are 5 causal variants per causal gene. When the effect size of causal variant on gene expression is large (e.g. Z = 5), 80% causal genes can be identified based on a cohort of 500 cases and 500 controls (**Fig 3**). The necessary sample size gradually increases as causal variants become less deleterious and have smaller functional consequence on gene expression, e.g. 2,000 samples will be needed to achieve 80% sensitivity for Z = 4.5. In contrast, given the same sample sizes standard GWAS will not have sufficient power to detect rare causal variants regardless of their effect sizes.

## Application to Alzheimer's disease

Alzheimer's disease is highly heritable, with heritability estimated to be as high as 60%-80% based on twin studies [20]. Large scale GWASs have identified multiple loci contributing to Alzheimer's disease, but the genetic variance explained by these loci is far below the level

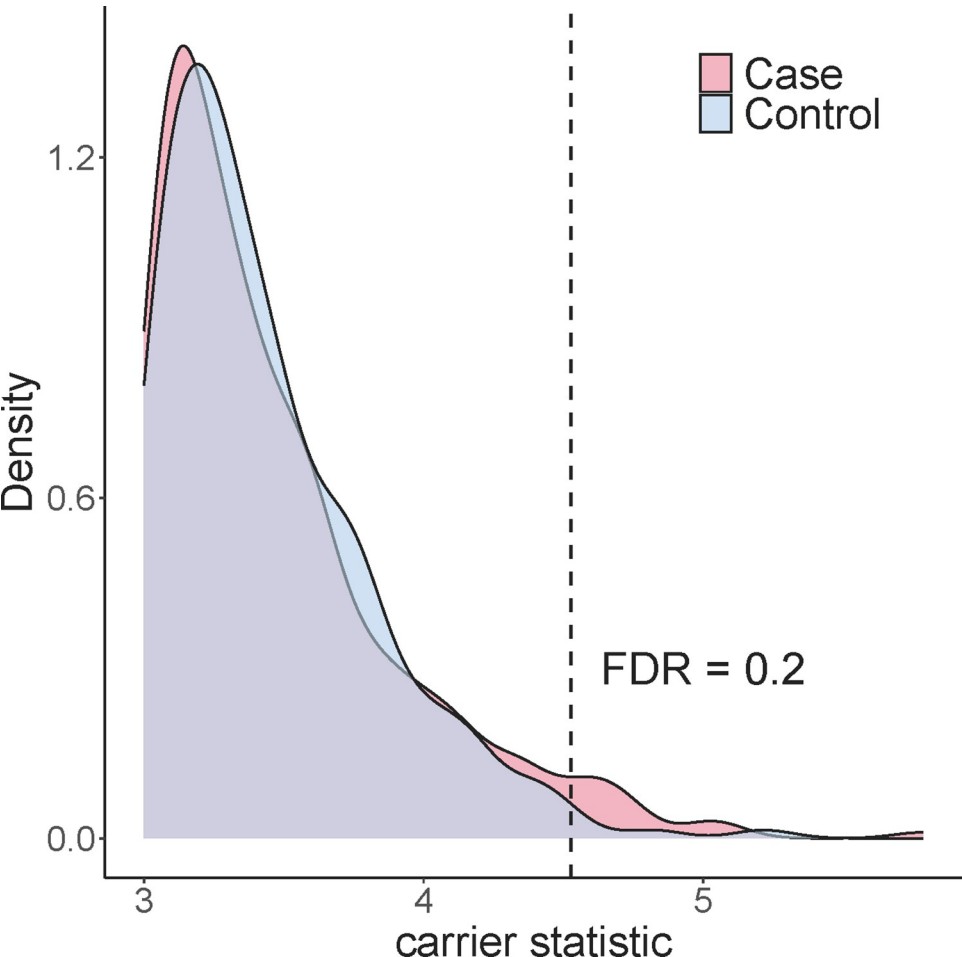

**Fig 4. Alzheimer's disease patients show significant excess of large carrier statistic.**

suggested by the disease heritability [21]. Additionally, there is limited understanding regarding the molecular mechanism through which these GWAS variants affect the disease, with the exception of the well-known APOE locus. To investigate whether rare variants (single nucleotide variants (SNVs) and short indels) confer functional consequences in Alzheimer's disease, we applied carrier statistic to a harmonized multi-omics dataset ($n_{case}$ = 444, $n_{control}$ = 234) consisting of WGS and RNA-seq from prefrontal cortex in four aging cohort studies: the Religious Orders Study (ROS) and Memory and Aging Project (MAP), the Mount Sinai Brain Bank (MSBB), and the Mayo Clinic (**Methods**). We found significant excess of large carrier statistic in the patients (**Fig 4**). Controlling FDR with a cutoff of 0.2, we prioritized 16 rare variants within 15 genes with large carrier statistic in the case group (**Table 1**), implicating them as candidate variants that may contribute to Alzheimer's disease through up-regulating gene expression in the brain. Carrier statistic adjusted for covariates (gender, age, and top 3 PCs of genotype) produces highly consistent results (**S4 Table**).

To see if existing methods can also detect these variants, we applied SKAT-O, coloc, and SAIGE-GENE+ to the same Alzheimer's disease dataset. These three methods did not identify any significant genes (FDR < 0.2), possibly due to insufficient sample size. To further evaluate the performance of carrier statistic, we assessed the enrichment of rare variants burden within the top prioritized genes in case group compared to that in controls. Among the top 100 genes

**Table 1. 16 rare variants within 15 genes with large carrier statistic in the Alzheimer's disease patients.**

| CHR | POS | REF | ALT | Variant | Gene | Carrier statistic |
|---|---|---|---|---|---|---|
| 14 | 31344166 | G | A | rs200935305 | COCH | 5.78 |
| 10 | 71716481 | G | A | rs537156091 | COL13A1 | 5.09 |
| 3 | 147108829 | T | TGTAGCCC | rs765482543 | ZIC4 | 5.02 |
| 15 | 63845640 | C | T | rs971269728 | USP3 | 5 |
| 15 | 81565527 | G | T | rs188062582 | IL16 | 4.77 |
| 10 | 115991333 | A | G | rs867202477 | TDRD1 | 4.74 |
| 6 | 37614984 | C | T | rs755089010 | MDGA1 | 4.71 |
| 1 | 29445783 | A | AT | rs1210817818 | EPB41 | 4.69 |
| 5 | 161113972 | A | G | rs966622465 | GABRA6 | 4.69 |
| 15 | 32907368 | GCCTTCAGC | G | rs1377672532 | ARHGAP11A | 4.62 |
| 7 | 31747403 | G | A | rs766182885 | PPP1R17 | 4.61 |
| 18 | 22804990 | G | A | rs770368967 | ZNF521 | 4.61 |
| 6 | 34028294 | C | T | rs556027021 | GRM4 | 4.58 |
| 1 | 231356659 | G | A | rs528099262 | TRIM67 | 4.56 |
| 6 | 37665148 | A | G | rs545009624 | MDGA1 | 4.55 |
| 6 | 123046921 | C | T | rs1016716974 | PKIB | 4.53 |

with largest carrier statistic, 67 genes have fold enrichment larger than 1, 32 genes have fold enrichment larger than 4/3, while only 6 genes have fold enrichment smaller than 3/4 (**S5 Fig**). Consistent with results from the simulations, enrichment of rare variants burden within each of those genes was moderate and did not pass significance threshold by the variants collapsing methods.

The significant genes prioritized by the carrier statistics may shed light on the genetic etiology of Alzheimer's disease (**Fig 5**). *COCH* has the largest carrier statistic of 5.78. Missense mutations within this gene were found to cause the late-onset DFNA9 deafness disorder [22,23]. Furthermore, deposits of Cochlin encoded by *COCH* is associated with age-related glaucomatous trabecular meshwork but absent in healthy controls. Additionally, SNPs inside *COCH* are associated with cortical thickness [24], changes in which, as ascertained through neuroimaging techniques, are commonly used in early detection and monitoring of

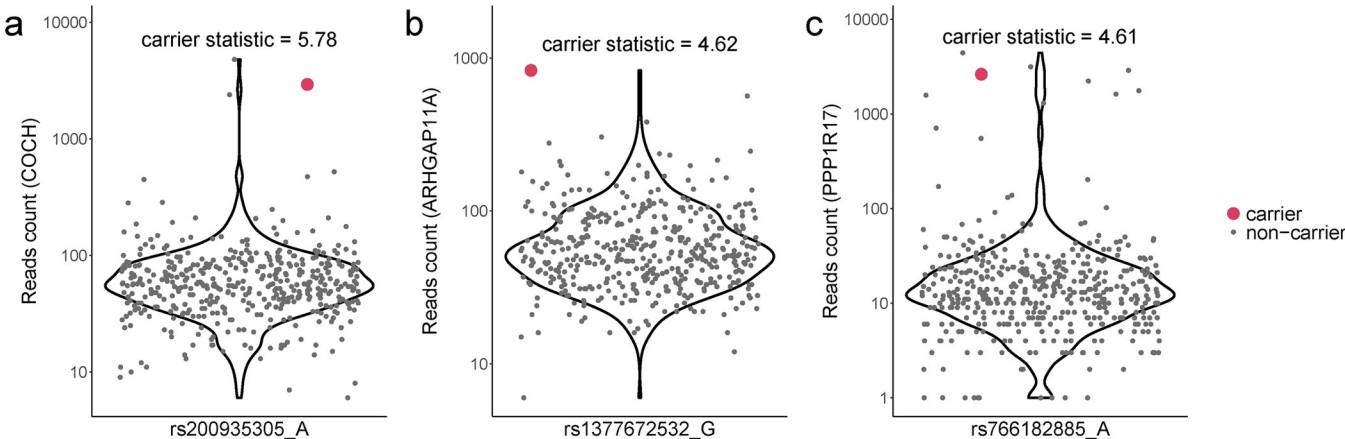

**Fig 5. Rare variant carriers show outlier expression level for (a) *COCH*, (b) *ARHGAP11A*, and (c) *PPP1R17*.** Pseudo count 1 was added to the RNA reads count for visualization purpose. Y-axis is on log scale.

Alzheimer's disease progression [25, 26]. Gene *ARHGAP11A* has a carrier statistic of 4.62. Transcribed mRNAs of the gene subcellularly localize and are locally translated in radial glia cells of human cerebral cortex and further regulate cortical development [27]. More importantly, *ARHGAP11A* may contribute to Alzheimer's disease pathology by mediating Amyloid-β generation and Amyloid-β oligomer neurotoxicity [28]. *PPP1R17* (carrier statistic = 4.61) functions as a suppressor of phosphatase complexes 1 (PP1) and 2A (PP2A). A recent study suggests that a subpopulation of neurons in the dorsomedial hypothalamus regulate aging and lifespan in mice through hypothalamic-adipose inter-tissue communication and that this regulation depends on Ppp1r17 expression [29]. Interestingly, *PPP1R17* is also involved in human-specific cortical neurodevelopment regulated by enhancers in human accelerated regions [30]. *ZIC4* (carrier statistic = 5.02) plays an important role in the embryonal development of the cerebellum. Heterozygous deletions encompassing the *ZIC4* locus are associated with a rare congenital cerebellar malformation known as the Dandy–Walker malformation [31]. Notably, mutations in proximity to the *ZIC4* locus are implicated in multiple system atrophy, a rare neurodegenerative disease [32]. A large scale GWAS of brain morphology has also identified associations with *ZIC4*, underscoring its significance in diverse neurological processes [33]. *MDGA1* (carrier statistic = 4.71) encodes a glycosylphosphatidylinositol (GPI)-anchored cell surface glycoprotein. It has been reported that *MDGA1* can contribute to cognitive deficits through altering inhibitory synapse development and transmission in the hippocampus [34]. Of note, *MDGA1* is one of the 96 genes from the Olink neurology panel with established links to neurobiological processes and neurological diseases.

## Discussion

We present carrier statistic, a statistical framework to perform multi-omics data analysis, for prioritization of disease-related rare variants and their regulated genes. Through simulations and analyses of real multi-omics datasets, we demonstrated that carrier statistic overcomes sample size limitations and achieves substantial gains in statistical power compared to existing variants collapsing methods. The superior performance of carrier statistic can be attributed to incorporation of functional gene expression data, which allows quantitatively measuring the impact of rare variants that cannot be determined by considering the variants on the DNA sequence level alone. Additionally, carrier statistic demonstrates robust performance irrespective of the number of underlying causal genes and in scenarios with unbalanced case-control ratios. We applied carrier statistic to Alzheimer's disease and highlighted several novel risk genes, providing insights into the molecular etiology of that complex disease.

It has been suggested that rare variants can be either deleterious or protective, and that variance component-based variants collapsing methods are more powerful than burden-type method for rare variants association studies, as they account for the different directions of variant effects [10,35]. While variance component-based methods evaluate the enrichment of deleterious variants burden in cases and enrichment of protective variants burden in controls, our method specifically assesses the enrichment of tail distribution of carrier statistics in cases. The rationale behind this approach is that deleterious variants are expected to significantly affect gene expression, whereas protective variants are more likely to maintain gene expression levels within a normal range. Consequently, extreme values for carrier statistics will be enriched in the case group, reflecting those rare deleterious variants that cause disease.

Carrier statistic serves as a general approach to study how rare variants affect complex disease through mediating gene expression. There exist several methods such as transcriptome-wide association study (TWAS) [36, 37] or colocalization [19, 38] that can also perform integrative analysis across multiple data modalities (genotype, gene expression, and phenotype).

However, those methods focus exclusively on effects of common SNPs and will have limited power for rare variants, as confirmed by the low sensitivity of coloc in both simulation studies and real application to Alzheimer's disease. In addition to SNVs and short indels that we included in this study, the statistical framework can be also applied to other types of rare variants (simple structural variants (SVs), complex SVs, mobile element insertions, tandem repeat expansions), which in general have larger effect size than SNVs [39]. Finally, carrier statistic can be adapted to other omics data, such as epigenomic and proteomic data.

In the real data application to Alzheimer's disease, we initially employed a traditional FDR cutoff of 0.05 and identified one significant gene, *COCH*, despite the constraints of a relatively small sample size consisting of 444 cases and 234 controls (totaling 678 individuals). Recognizing the inherent limitations in statistical power with this sample size, we subsequently increased the FDR cutoff to 0.2. This allowed us to enhance sensitivity and detect a broader set of potential gene associations, identifying a total of 15 genes that may be linked to Alzheimer's disease. In future applications to diseases with larger sample size, we recommend switching back to the traditional FDR cutoff of 0.05, which can minimize false positives and ensure robustness in scientific findings.

Over the last fifteen years, abundant disease-associated loci have been identified based on genome sequences of biobank-scale sample size (e.g. hundreds of thousand) [40,41]. However, even with such large sample sizes it is still difficult for GWAS analysis to detect rare causal variants. Another important objective is to understand the biological roles of the detected loci, which remains challenging. We showed here that by integrating RNA-seq data, the carrier statistic approach offers a study design that may overcome the sample size limitation and may help to associate the functional rare variants and their target genes. This approach will be especially useful for the study of diseases for which biospecimen of disease relevant tissue are easy to obtain and RNA-seq can be performed, such as autoimmune disease (relevant to blood) or skin-related disease. Fortunately, there already exist several large datasets that satisfy this condition, including the Genotype-Tissue Expression project (GTEx) [42], The Cancer Genome Atlas program (TCGA) [43], and PsychENCODE [44]. The Trans-Omics for Precision Medicine program (TOPMed) [45] is another significant resource, which has already produced over 100K WGS samples and is generating a comprehensive range of multi-omics data (RNA-seq, metabolomics profiling, DNA methylation profiling, and proteomics assay). These extensive datasets provide ample opportunities for applying our method and analyzing large-scale multi-omics data. We hope that the results presented in this paper will serve to demonstrate the promise of the carrier statistic approach and will stimulate interest in the collection of both genotype and gene expression data for the same individuals in future case-control studies. Notably, when the tissue sample is available, adding RNA-seq to a WGS-based GWAS will not increase the cost of the study significantly. As multi-omics data accumulates alongside genome sequencing data, we anticipate that carrier statistics will become an effective approach to dissect the molecular mechanism of complex diseases.

## Methods

### Carrier statistic

For each rare variant-gene pair (the variant is located within the exon of the gene), we used the expression of that gene in the rare variant noncarriers as the null distribution and computed a z-score for each rare variant carrier, then average over carriers of that variant. To formally define this concept, for each rare variant-gene pair, let $E_i^{carrier}(i = 1, \ldots, n^{carrier})$ and $E_j^{noncarrier}(j = 1, \ldots, n^{noncarrier})$ represent the gene expression levels for carriers and noncarriers of that rare variant, respectively. Let $m^{noncarrier}$ and $\sigma^{noncarrier}$ denote the mean and standard

deviation of gene expression among all rare variant noncarriers. Then the expression association z-score is computed as $\frac{1}{n^{carrier}} \sum_{i=1}^{n^{carrier}} \frac{E_i^{carrier} - m^{noncarrier}}{\sigma^{noncarrier}}$, referred to hereafter as carrier statistic. The carrier statistic was computed separately within the case group and the control group. Rare variants were defined as SNVs and short indels whose allele count was no larger than 5 within the case group or within the control group. Therefore, the rare variants and thus the number of carrier statistics are not the same between two groups. The carrier statistic can be interpreted as expression effect size which quantifies the degree to which the rare variant impacts gene expression level. We assume that diseased population shows enrichment in rare variants that have large impact on expression for disease-related genes, thus there will also be enrichment of extreme carrier statistic for disease-related rare variant-gene pairs in the case group. We prioritize those rare variants and genes with outlier carrier statistic in the case group. For rare variant-gene pairs with positive carrier statistic, false discovery rate at a given threshold of carrier statistic, denoted by $z_0$, can be computed as $\frac{\Pr[z_{ctrl} \geq z_0]}{\Pr[z_{case} \geq z_0]}$, where $z_{case}$ and $z_{ctrl}$ denote the carrier statistic in the case group and control group, respectively. Duplicative carrier statistics were removed (i.e. multiple rare variants occurring in the same individuals are counted as the same rare variant). Similarly, for rare variant-gene pairs with negative carrier statistic, false discovery rate at threshold of $z_0$ can be computed as $\frac{\Pr[z_{ctrl} \leq z_0]}{\Pr[z_{case} \leq z_0]}$.

## Simulations

We simulated genotypes for a large population consisting of 125,748 individuals based on the alternative allele count from 125,748 exomes in the gnomAD v2.1.1 dataset [2]. Only exonic variants that passed all variant filters in the gnomAD dataset were retained. Then we simulated gene expression data for the large population as follows. We first simulated background gene expression profile for these 125,748 individuals while matching the mean and standard deviation of normalized gene expression in the reference expression dataset. In this study, we used $\log_2$(reads count+1) as normalized gene expression and RNA-seq in the whole blood tissue from GTEx project v8 as the reference expression dataset [42]. Genes whose median number of reads count in the large population < 10 were removed. We randomly selected $m$ causal genes and $l$ causal variants for each causal gene, where $m$ was set as 50 and $l$ was set to vary from 1 to 10. The minor allele count (MAC) for rare causal variants spans from 1 to 125, with the corresponding MAF ranging from 0.0004% to 0.05%. For each causal gene, we perturbed the gene expression for causal variant carriers by $z*sd$ fold, where $z$ was set as 5 and $sd$ was the standard deviation of normalized gene expression in the reference dataset. Next, we simulated the disease status for the large population by assuming penetrance of causal variant as $p_{carrier}$ and prevalence in causal variant noncarrier as $p_{noncarrier}$. Here $p_{carrier}$ varied from 0.5 to 0.9 and $p_{noncarrier}$ varied from 0.005 to 0.02. Finally, we randomly sampled 500 cases and 500 controls from the affected and nonaffected population respectively to mimic the sample recruitment procedure in the disease study. Each simulation setting was repeated for 100 times. We also assessed the performance of carrier statistic when only one causal gene was present and case-control ratio was highly unbalanced (500 cases and 50,000 controls, reflecting a 1% case-control ratio).

We evaluated the performance of different methods using two metrics: FDR and sensitivity. FDR was defined as the proportion of falsely identified genes among all identified ones. If no gene was identified then FDR was set as 0. Sensitivity was defined as the proportion of truly identified genes among all underlying causal genes.

Note that carrier statistic was computed by using gene expression from rare variant noncarriers in the same group (i.e. case or control) as the carriers as null distribution. We also

checked if using gene expression from noncarriers in both case group and control group as the null distribution will produce false positive findings. We perturbed expression level for all genes in the case group. In this case, FDR showed substantial inflation for using gene expression from all individuals in two groups as null distribution, especially when there is large systematic difference in the transcriptome between two groups (S2B Fig). On the contrary, using gene expression from rare variant noncarriers in the same group of carriers as null distribution consistently controlled FDR at the nominal level (S2A Fig).

While it is possible that rare protective variants exist in the control group, we assume they would typically regulate gene expression level within a normal range rather than causing significant alterations. To further address this, we conducted additional simulations to evaluate the performance of carrier statistic in the presence of both deleterious, rare causal variants and rare protective variants. In these simulations, we randomly selected 50 causal genes and 50 protective genes, each containing 5 rare causal variants or 5 protective variants. Gene expression among rare variant carriers was perturbed by $z*sd$ fold, where $z$ was set to 5 for rare causal variants and 2.5 for rare protective variants, with $sd$ representing the standard deviation of normalized gene expression in the reference dataset. The penetrance of causal variant was set as 0.7. Genotypic relative risk for protective variants was set as 0.1 (presence of a protective variant will decrease disease risk by tenfold). Prevalence for non-carriers of both types was set as 0.01. Due to low disease prevalence, protective variants are unlikely to be observed in the control group, and their presence in the case group is even less probable as protective variants imply a lower risk of disease. Therefore, we adjusted the population MAF for protective variants to ensure their representation in the control group. Specifically, setting the population MAF for protective variants to 0.1% led to an observed frequency in the control group similar to that of causal variants in the case group (S5 Table). Under these simulation conditions, carrier statistic still effectively controlled FDR at the nominal level (S3A Fig) and achieved over 75% sensitivity in identifying causal genes (S3B Fig).

## Implementation of different methods

SKAT-O was performed using the R package SKAT v.2.2.5. SAIGE-GENE+ was performed using SAIGE v1.1.9. coloc was performed using the R package coloc v.5.2.3. Genes with a posterior probability of colocalization larger than 0.1 were identified as potentially related to the disease. Here, we used this lower threshold of 0.1 instead of the default 0.5 to increase sensitivity of the coloc method. Both common and rare variants were included in the analysis.

## Multi-omics data analysis for Alzheimer's disease

WGS data for the four aging cohorts (ROS/MAP, MSBB, and the Mayo Clinic) were obtained from the Whole Genome Sequence Harmonization Study (Synapse ID: syn22264775). RNA-seq data for the same four cohorts were downloaded from the RNAseq Harmonization Study (syn21241740). Only white people with both WGS and RNA-seq data were included in the analysis.

We determined disease status following description in previous publications [46,47]. For the ROS/MAP cohorts, individuals with a Braak neurofibrillary tangle score $\geq 4$, a CERAD neuritic and cortical plaque score $\leq 2$, and a cognitive diagnosis of probable Alzheimer's disease with no other causes (cogdx = 4) were classified as cases, while individuals with a Braak score $\leq 3$, a CERAD score $\geq 3$, and a cognitive diagnosis of no cognitive impairment (cogdx = 1) were classified as controls. For MSBB, individuals with a Braak score $\geq 4$, a CERAD score $\geq 2$, and a Clinical Dementia Rating (CDR) score $\geq 1$ were classified as cases, while individuals with a Braak score $\leq 3$, a CERAD score $\leq 1$, and a CDR score $\leq 0.5$ were

classified as controls. For the Mayo Clinic cohort, individuals with a Braak score $\geq 4$ and a CERAD score $\geq 2$ were classified as cases, while individuals with a Braak score $\leq 3$ and a CERAD score $\leq 1$ were classified as controls. Of note, definition of CERAD score in the ROS/MAP cohort is different from that in the MSBB and the Mayo Clinic cohorts. After harmonization across cohorts, 444 cases and 234 controls in total were identified and used for downstream analysis.

Next, we performed quality control on the WGS and RNA-seq data. For WGS data, only exonic variants with missing genotypes < 10% were retained. For RNA-seq data, we selected prefrontal cortex as the target brain tissue. If a donor does not have RNA-seq in the prefrontal cortex, then RNA-seq in other tissues will be used based on the following order: dorsolateral prefrontal cortex > posterior cingulate cortex > head of caudate nucleus in the ROS/MAP cohort, prefrontal cortex > frontal pole > superior temporal gyrus > inferior frontal gyrus > parahippocampal gyrus in the MSBB cohort, and temporal cortex > cerebellum in the Mayo Clinic cohort. Genes with zero reads count in more than 10% of samples or with median reads count < 10 across samples were excluded. $\log_2$(reads count+1) was used as normalized gene expression. Then we applied carrier statistic to perform downstream analysis. Additionally, we adjusted gene expression for covariates, including age, gender, and top 3 PCs of genotype, separately within the case and control groups. Carrier statistic, both with and without covariates adjustment, gave highly consistent results (**Table 1** and **S4 Table**).

## Supporting information

**S1 Fig. FDR for carrier statistic, SKAT-O, coloc, and SAIGE-GENE+ in simulations with varying (a) penetrance of causal variant, (b) prevalence in causal variant noncarriers, and (c) number of causal variants per causal gene.** Error bar shows standard deviation across 100 simulation repeats.
(TIF)

**S2 Fig. (a). FDR is well-calibrated for using gene expression from noncarriers in the same group of carriers as null distribution. (b). FDR showed substantial inflation for using gene expression from all noncarriers as null distribution.** Z quantifies the level of systematic difference in the transcriptome between case group and control group. Error bar shows standard deviation across 100 simulation repeats.
(TIF)

**S3 Fig. (a) FDR and (b) sensitivity of carrier statistic in simulations where both rare causal variants and rare protective variants are present.** Error bar shows standard deviation across 100 simulation repeats.
(TIF)

**S4 Fig. (a) FDR and (b) sensitivity of carrier statistic, SKAT-O, coloc, and SAIGE-GENE + in simulations with only 1 causal gene and 1% case-control ratio.** Error bar shows standard deviation across 100 simulation repeats.
(TIF)

**S5 Fig. Enrichment of rare variants within top 100 genes with largest carrier statistic in Alzheimer's disease patients.** Genes were ranked according to decreasing order of carrier statistic. Fold enrichment was defined as the ratio of rare variants burden within the gene in case group compared to that in the control group.
(TIF)

**S1 Table. Rare variants burden in simulated disease model with varying penetrance of causal variant.**
(XLSX)

**S2 Table. Rare variants burden in simulated disease model with varying prevalence in causal variant noncarriers.**
(XLSX)

**S3 Table. Rare variants burden in simulated disease model with varying number of causal variants per causal gene.**
(XLSX)

**S4 Table. 17 rare variants within 16 genes with large carrier statistic adjusted for covariates in the Alzheimer's disease patients.**
(XLSX)

**S5 Table. Rare variants burden in simulated case-control data when population MAF of protective variants is 0.1%.**
(XLSX)

## Acknowledgments

We thank Dr. Bo Zhou and Dr. Hua Tang for discussion and providing feedbacks.

## Author Contributions

**Conceptualization:** Hanmin Guo, Wing Hung Wong.

**Data curation:** Hanmin Guo.

**Formal analysis:** Hanmin Guo.

**Funding acquisition:** Alexander Eckehart Urban, Wing Hung Wong.

**Investigation:** Hanmin Guo, Alexander Eckehart Urban, Wing Hung Wong.

**Methodology:** Hanmin Guo, Wing Hung Wong.

**Project administration:** Hanmin Guo, Alexander Eckehart Urban, Wing Hung Wong.

**Supervision:** Alexander Eckehart Urban, Wing Hung Wong.

**Validation:** Hanmin Guo, Alexander Eckehart Urban, Wing Hung Wong.

**Visualization:** Hanmin Guo.

**Writing – original draft:** Hanmin Guo.

**Writing – review & editing:** Hanmin Guo, Alexander Eckehart Urban, Wing Hung Wong.

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
