## [Decision Letter · Decision Letter 0]

8 Jul 2024

Dear Dr Wong,

Thank you very much for submitting your Research Article entitled 'Prioritizing disease-related rare variants by integrating gene expression data' to PLOS Genetics.

The manuscript was fully evaluated at the editorial level and by independent peer reviewers. The reviewers appreciated the attention to an important problem, but raised some concerns about the current manuscript. Based on the reviews, we will not be able to accept this version of the manuscript, but we would be willing to review a much-revised version. We cannot, of course, promise publication at that time.

If you decide to revise the manuscript for further consideration at PLOS Genetics, please aim to resubmit within the next 60 days, unless it will take extra time to address the concerns of the reviewers, in which case we would appreciate an expected resubmission date by email to plosgenetics@plos.org.

We are sorry that we cannot be more positive about your manuscript at this stage. Please do not hesitate to contact us if you have any concerns or questions.

Yours sincerely,

Hongyu Zhao

Guest Editor

PLOS Genetics

Xiaofeng Zhu

Section Editor

PLOS Genetics

Reviewer's Responses to Questions

**Comments to the Authors:**

Reviewer #1: Guo et al. proposed a carrier statistic to prioritize rare variants using gene expression data, with limited sample size. While it is a great attempt to solve a critical problem, there are a number of limitations.

1. The method needs more details. It is unclear how to compute the z score for each rare variant carrier. I assume carrier means an individual carrying the rare variant. I suggest write the definition using math formula explicitly. Similarly, the caption for Figure 1 requires elaboration, as the x-axis in Figure 1a is not clearly defined.

2. The method requires both genotypes and gene expression for the same samples. This is very hard to achieve in practice, as the genotyped samples often differ from those available expression data. So the method application is very limited.

3. The authors compared the methods with burden test, SKAT and SKAT-O, which do not incorporate expression information. So the comparison is unfair. It would be beneficial to compare the method with others that also utilize expression data. What about colocalization or similar methods?

4. In the Alzheimer application, the authors used FDR cutoff 0.2, which is quite high. Clarification on whether this threshold is recommended for the method, or if it was specifically chosen for this application, would be valuable.

Reviewer #2: The review is uploaded as an attachment.

**Have all data underlying the figures and results presented in the manuscript been provided?**

Reviewer #1: **No: **The method code and the simulation, application code are not available.

Reviewer #2: Yes

PLOS authors have the option to publish the peer review history of their article (what does this mean?). If published, this will include your full peer review and any attached files.

Reviewer #1: No

Reviewer #2: No

---

## [Decision Letter · Decision Letter 1]

29 Aug 2024

Dear Dr Wong,

We are pleased to inform you that your manuscript entitled "Prioritizing disease-related rare variants by integrating gene expression data" has been editorially accepted for publication in PLOS Genetics. Congratulations!

Yours sincerely,

Hongyu Zhao

Guest Editor

PLOS Genetics

Xiaofeng Zhu

Section Editor

PLOS Genetics

Comments from the reviewers (if applicable):

Reviewer's Responses to Questions

**Comments to the Authors:**

Reviewer #1: Thanks for the detailed response to the issues. I have some follow-up questions regarding the carrier statistic. Could you control for covariates (age, sex, PCs)? It appears this was not done. Could you comment on the impact of not controlling covariates? Additionally, is there any interpretation of the carrier statistic? Does it have any meaningful implications, and can it be interpreted as expression effect size?

Reviewer #2: I appreciate the authors’ efforts for making the improvement of the manuscript, especially adding the mathematical definition and additional simulations, and comparisons with coloc and SAIGE-GENE+. The updated version of the manuscript and the response letter properly addressed my previous questions/remarks.

**Have all data underlying the figures and results presented in the manuscript been provided?**

Reviewer #1: Yes

Reviewer #2: Yes

PLOS authors have the option to publish the peer review history of their article (what does this mean?). If published, this will include your full peer review and any attached files.

Reviewer #1: No

Reviewer #2: No

**Data Deposition**

http://datadryad.org/submit?journalID=pgenetics&manu=PGENETICS-D-24-00550R1

**Press Queries**

---

## [Editor Report · Acceptance letter]

24 Sep 2024

PGENETICS-D-24-00550R1 

Prioritizing disease-related rare variants by integrating gene expression data 

Dear Dr Guo, 

We are pleased to inform you that your manuscript entitled "Prioritizing disease-related rare variants by integrating gene expression data" has been formally accepted for publication in PLOS Genetics! Your manuscript is now with our production department and you will be notified of the publication date in due course.

With kind regards,

Anita Estes

PLOS Genetics

On behalf of:
